# Effect of Shear Bands Induced by Asymmetric Rolling on Microstructure and Texture Evolution of Non-Oriented 3.3% Si Steel

**DOI:** 10.3390/ma13214696

**Published:** 2020-10-22

**Authors:** Zhiyong Fang, Yanhui Guo, Bin Fu, Liqun Wei, Jun Chen, Linghuan Pang, Zemin Wang

**Affiliations:** School of Materials Science and Engineering, Shanghai Institute of Technology, Shanghai 201418, China; 186081104@mail.sit.edu.cn (Z.F.); weiliqun@sit.edu.cn (L.W.); 196082109@mail.sit.edu.cn (J.C.); plh9009@sit.edu.cn (L.P.); wzm@sit.edu.cn (Z.W.)

**Keywords:** non-oriented silicon steel, asymmetric rolling, microstructure and texture, Goss, shear band

## Abstract

In the present work, the microstructure and texture of non-oriented 3.3% Si steel processed by asymmetric rolling (ASR) and subsequent annealing at different temperatures were compared with those obtained when using traditional symmetric rolling (SR). This work aims to reveal the effect of shear bands introduced by the ASR on the microstructure and texture evolution. The ASR sample reaches a recrystallization fraction of 62% at an annealing temperature of 650 °C, which is 32% higher than that of the SR sample annealed at the same temperature. This can be attributed to the abundant shear bands introduced by the ASR, which serve as the heterogeneous nucleation sites for the recrystallized grains. When increasing the annealing temperature to 750 °C, complete recrystallization could be observed in both asymmetric- and symmetric-rolled samples. When using an annealing temperature of 650 °C, the γ-oriented grains were dominant in the surface layer, while strong Goss-oriented grains could be observed in the center in the ASR sample. This is due to the fragmented small subgrains with different orientations in the surface layer inhibiting the nucleation of Goss- and cube-oriented grains during the annealing. In contrast, numerous Goss- and cube-oriented grains were formed in the surface layer after complete recrystallization when the ASR sample was annealed at a temperature of 750 °C. This may be related to the higher thermal energy, which benefits the nucleation of the Goss- and cube-oriented grains. In addition, ASR significantly increased the strength of η-fiber after complete recrystallization when compared with SR. This work might be helpful to design the rolling and the subsequent annealing processes.

## 1. Introduction

Non-oriented silicon steel has excellent magnetic properties and is widely used in industries [1,2,3]. Researchers aim for the continuous improvement of the magnetic properties of non-oriented silicon steel. Generally, magnetic induction is mainly dependent on the texture, while the core loss, the sum of hysteresis loss and the eddy current loss, are minimized at the optimum grain size. Consequently, the recrystallization behavior during the annealing process can significantly affect the magnetic properties of non-oriented silicon steel [4,5,6,7,8]. There are two common methods to improve the magnetic properties of materials [9]. The first one is to fine-tune the alloy composition of the material and introduce new precipitation particles to control the evolution of the texture during the recrystallization process [10,11]. The other and currently most common method is to optimize the microstructure and texture by processing technology, such as twin roll casting or rolling [12,13].

Studies have shown that the recrystallized grains preferentially nucleate at the shear bands [8,14], whose orientation directly controls the texture evolution during the recrystallization. Goss grains are mainly nucleated in the deformed grains with the {111} <112>, {111} <110> and {112} <110> orientations within the shear bands. The nucleation of new cube grains also takes place within shear bands, but the density of nuclei is lower than that of Goss grains [15,16]. Therefore, shear bands are closely related to these textures, which improve the magnetic properties of the material after annealing. It has been reported that asymmetric rolling process (ASR) is an efficient way to introduce shear deformation, which induces the formation of shear bands in the material, compared with the conventional symmetric rolling (SR) [17,18].

In ASR, the speeds of the two rollers are different, and hence strong shear deformation is formed across the entire sheet thickness [19,20,21]. Logically, this shear deformation is asymmetrically distributed in the material and differs from that imposed by conventional symmetric rolling (SR). Besides the symmetry, the effective plastic strains reached by the ASR are larger than those obtained by the SR at the same rolling reduction [22,23,24,25,26]. In recent years, many studies have been performed on ASR of alloys and steel [22,27,28]. The researchers pointed out that ASR can produce more severe shear deformation, which is beneficial to refine the recrystallized grains [23,29]_._ With respect to the magnetic properties, silicon (Si) steel provides one example where the ASR can simultaneously increase the magnetic induction and decrease the iron loss with a maximum amplitude of 0.011 T and 0.50 W/kg [21]. Silva et al. [30] also found that the ASR Si steel exhibits significantly enhanced magnetic properties compared with the SR one under the same recrystallization condition. However, fewer studies have been conducted on the effect of shear bands on the microstructure and texture evolution of non-oriented silicon steel.

For the present work, Si steel was asymmetrically and symmetrically rolled with subsequent annealing at different temperatures. The optical microscope (OM) and the electron backscatter diffraction (EBSD) were used to characterize the microstructure and texture of the samples. The main proposal of this work is to understand the mechanism of the microstructure and texture evolution during the annealing from the viewpoint of shear bands. The results of this work might be helpful to design the rolling and subsequent annealing processes.

## 2. Experimental Procedure

The as-received sheet has a thickness of 2 mm and a chemical composition of 0.006% C, 3.3% Si, 0.2% Mn, 0.03% P and 0.006% S (in mass percent). The sheet was asymmetrically cold-rolled to 0.5 mm by 0.08 mm reduction per pass. The upper and lower rollers with the equivalent diameters of 180 mm had velocities of 13 turns/min and 10 turns/min, respectively. For comparison, the SR was also performed with the same setup, and the roller speed was 10 turns/min. Samples were cut from the asymmetrically and symmetrically rolled sheets to be used as references.

The schematic illustration and the finite element analysis (FEA) of the ASR are presented in Figure 1. The simulations were carried out by using the software MSC Marc. The setup of the FEA speed ratio was equivalent to the real rolling, and the rolling reduction was set to 20%. The deformation zone was the area between the entrance and the exit. The difference in rotational speed between upper and lower rollers caused an asymmetric deformation of the sheet, as shown in Figure 1b, which stayed in good agreement with [18]. Microscopically, the grid distortions were also asymmetric (see the asymmetric deformed grids in Figure 1b). Due to the asymmetric deformation, the neutral point at the upper interface deviated from that of the lower interface in the rolling direction (RD), as shown by the yellow arrows in Figure 1b. This heterogeneously distributed shear deformation may lead to the formation of shear bands comprising fine sub-grains [8].

The cold-rolled steel strips were annealed in a box electric furnace in temperature range from 600 to 850 °C with interval of 50 °C for 5 min. After annealing, samples were cut from the sheets and manually polished and etched with 4% nital solution (96% alcohol + 4% nitric acid). Then, the microstructures were observed using an OM. Note that the observed plane was the cross-section between the RD and normal direction (ND). The recrystallized fractions and the grain size were determined by the software Image J. The EBSD analysis was performed on the cross-sections, which covered the entire thickness of the sheets. The measurement data from EBSD were processed by orientation imaging microscopy (OIM) and channel 5 analysis software to obtain data such as the recrystallization fraction, the entire thickness orientation distribution function (ODF) figures and the ODF figures of different layers. Here, a parameter S, which is defined as S = 2 d/t, where t and d are the thickness and the distance from the measuring layer to the center layer, respectively, is used to denote the observed layers. Accordingly, S = 1 represents the near surface layer with the upper roller, and S = 0 represents the center layer. The evolution of the main textures such as {001} <110>, {112} <110>, {111} <110>, {111} <112>, {011} <001> (Goss) and {001} <100> (cube) were monitored. The first four orientations are usually observed in the cold-rolled and annealed texture of steel. The last orientation is preferred for the magnetic properties of non-oriented electrical steel because it has an easy magnetization direction of [100] in a plane parallel to the surface of the sheet [31,32].

## 3. Results

### 3.1. Microstructure

#### 3.1.1. Microstructure after Rolling without Annealing

Figure 2 presents the optical microstructure of the RD–ND plane covering the entire thickness of the sheets directly after rolling (without annealing). Obviously, two different deformation areas, which are denoted as regions A and B in Figure 2a and regions C and D in Figure 2b, can be observed. The degree of etching has a significant relationship with the energy it contains [15,16]. The regions A and C have high energy, which is related to the rich defects such as shear bands or dislocations that they contain, and are easy to etch, resulting in the dark area in the OM image. In contrast, the regions B and D contain low energy and are relatively difficult to etch, so the area appears bright. The distributions of the two types of deformation areas are different in the SR and ASR samples. In the ASR sample, the dark etching area is more homogeneously distributed along the ND direction. Furthermore, more shear bands, which are indicated by the arrows, can be observed in the ASR sample. Note that the shear bands in the ASR sample exhibit different orientation compared with those in the SR sample. The shear bands in the ASR sample have a larger inclination angle around 40° with the RD, while the inclination angle between the shear bands and RD in the SR sample amounts to an approximate value of 30°. This observation is in accordance with the results of [33].

#### 3.1.2. Microstructure after Annealing

Figure 3 shows the optical microstructure covering the entire thickness of the sheets after annealing. After annealing at 600 °C, no recrystallization nuclei are generated in either the ASR or SR samples, indicating that the samples might be still in the recovery stage. Upon increasing the annealing temperature, nucleation and subsequent growth of the recrystallized grains can be clearly observed. It should be mentioned that the size of the recrystallized grains increases and the grain size distribution gradually becomes uniform with temperature increase. Quantitatively, the average grain size increases from 31.4 μm (20.5 μm) at 750 °C to 60.4 μm (62.5 μm) at 850 °C for the ASR (SR) sample. The mean square deviation of the ASR sample at 850 °C amounts to 16.6 μm, which is lower than that of the SR sample (23.5 μm), indicating a more uniform grain size distribution in the ASR sample. Note that the 750 °C is the temperature at which a complete recrystallization is achieved in both samples.

Figure 4 represents the recrystallization fraction as a function of the annealing temperature. At an annealing temperature of 650 °C, the ASR sample exhibits a recrystallization fraction of 62%, which is significantly higher than that of the SR sample with a fraction of 30%. Both in the ASR and SR samples, the recrystallization fraction increases with the annealing temperature increasing. In addition, at higher temperatures, the rate of increase of the recrystallization fraction decreases in the ASR sample, while the SR sample shows a relatively constant rate (see the slopes of the ASR and SR curves in Figure 4). Furthermore, it is well known that a longer annealing time is beneficial to the recrystallization, i.e., the longer the annealing time, the higher the recrystallization fraction. However, here we mainly focus on the temperature effect rather than the time effect.

### 3.2. Texture

#### 3.2.1. Texture after Rolling without Annealing

Figure 5 shows the ODF figures of the entire thickness after the ASR and SR. After the SR, the textures show a strong α-fiber (<110>//RD) with a peak at {112} <110> and relatively weak λ- (<100>//ND) and γ-fibers (<111>//ND). However, after the ASR, the λ- and γ-fibers become significantly stronger, and the {112} <110> component is still strongest, but weaker than that of the SR sample. Previous studies have found that the shear bands consisting of strong {111} <110> and {111} <112>-oriented grains are preferential nucleation sites for Goss texture during recrystallization [14,15,34]. Therefore, ASR is expected to produce a stronger Goss texture during recrystallization. To further study the effect of asymmetric rolling on the texture along the thickness of the sample, the orientation distribution function (ODF) figures of different layers are also compared, as shown in Figure 6. After the ASR, the strongest {001} <110> and the second strongest γ-fiber with orientations such as {111} <110> develop at the surface. However, in the other layers, the strength of the {112} <110> orientation is significantly enhanced, and the strengths of the {001} <110> and {111} <110> orientations dramatically weaken. As a whole, the texture components of the center layer and the subsurface layer are similar. However, the texture of the center layer differs from that of the surface in that a strong γ-fiber texture appears at the surface layer, while it is relatively weak in the center layer.

#### 3.2.2. Texture during Recrystallization

ODF images of the entire thickness of the samples after annealing at 650 and 750 °C (ASR) are demonstrated in Figure 7. Partial recrystallization occurs at an annealing temperature of 650 °C. The γ-fiber (mainly {111} <110> and {111} <112>) is reduced, while the Goss texture is enhanced. The strength of {001} <110> orientation does not change. Regarding the previous research results [15,31] and the data from this study, it is suggested that Goss-oriented grains nucleate within the γ-fiber grains or at the grain boundaries. The Goss-oriented grains grow by consuming the γ-fiber-oriented grains. The strengths of the {001} <110>-oriented grains of the ASR samples do not change after annealing at a temperature of 650 °C (see Figure 5b and Figure 7a), indicating these grains are stable. Increasing the annealing temperature to 750 °C, the grains with the {100} <110> orientation are exhausted (see Figure 7a,b). At the same time, the cube orientation appears in the ODF, while the strength of the Goss orientation does not significantly change, as shown in Figure 7b.

Figure 8 shows the ODF of different layers in the ASR sample after annealing. After annealing at 650 °C, the surface of the sample shows a strong γ-fiber, as shown in the third subfigure (from left to right) in Figure 8a. In the subsurface layer, the {112} <110> component is completely consumed as a result of the cold rolling (see the second subfigures in Figure 6 and Figure 8a), but a strong {001} <110> component, a relatively weak γ-fiber and Goss texture are available. In contrast, in the center layer, the α-fiber produced by the cold rolling is relatively stable, and this texture is almost completely retained, as shown in the first subfigure in Figure 8a. Comparing the first subfigure in Figure 6 and the first subfigure in Figure 8a, it can be observed that the γ-fiber is almost completely consumed, while the Goss texture strength is significantly enhanced. When the temperature is raised to 750 °C, the surface is dominated by η-fiber ({100} <001> and {110} <001>) texture, as shown in the third subfigure in Figure 8b. The subsurface layer is mainly dominated by γ- and λ-fibers (see the second subfigure in Figure 8b).

#### 3.2.3. Texture Comparison after Complete Recrystallization

Figure 9 shows the texture distribution of the entire thickness of the SR and ASR samples after annealing at 850 °C. Figure 10 compares the volume fraction of four typical texture fibers measured by channel 5 analysis software for the SR and ASR samples, with a maximum deviation of 20° from each ideal orientation. The η-fiber fraction in the SR sample amounts to 24.5%, while in the ASR sample it is 33.4%. Meanwhile, the γ-fiber fraction in the ASR sample is 6.2% lower than that in the SR sample. It can be concluded that the ASR can increase the strength of η-fibers, which is beneficial in improving the magnetic properties of non-oriented Si steel.

## 4. Discussion

### 4.1. Effect of Rolling Mode on the Microstructure after Rolling and Annealing

Compared with the SR, ASR will induce additional shear stress, which will cause the steel sheets to undergo more severe shear deformation [31]. Currently, although the fundamental theory regarding the formation of shear bands is still under debate, it can be generally accepted that shear band is caused by shear deformation on specific shear systems, whether it is crystallographic [32] or non-crystallographic [35]. In this context, ASR can produce more shear bands compared with SR [29,30]. At the same time, it has been confirmed that the difference in etching degree is related to the stored energy of the deformation bands after cold rolling [15]. The higher the stored energy, the easier the etching. Therefore, we can observe that the shear bands tend to distribute at the surface and subsurface of the steel sheets after symmetric rolling, such as region A in Figure 2, while they are also available in the center layer after ASR, such as region C in Figure 2. It is well known that shear bands can provide more nucleation positions for new crystallized grains [15,36,37]. Thus, when the annealing temperature is 650 °C, numerous nucleation positions are preferentially provided in the ASR sample containing more shear bands rather than the SR sample, where the bright-etched elongated grains with lower stored energy retain. Therefore, the recrystallization fraction increase rate of the ASR sample is much higher than that of the SR sample after annealing at 650 °C, as shown in Figure 4. At higher temperatures, the increase of recrystallization fraction decelerates in the ASR sample, while in the SR sample, this rate roughly remains as a constant. This may be related to the lower driving force (P) of the bright-etched regions of the ASR sample. In physical metallurgy, the migration velocity (V) of a moving interface is generally considered to be the product of mobility (M) and driving force (P), i.e., V = M∙P [38]. Because the driving force (P) of the remaining bright-etching region of the ASR sample is low, a larger mobility (M) is needed to achieve the migration speed of the mobile interface. Note that the mobility (M) is exponentially dependent of the temperature. In addition, Figure 11 shows the microstructure and texture of asymmetric-rolled sample after annealing at 650 °C. We find that the deformation zones without nucleation are mainly {100} <110>- and {112} <110>-orientated. It has been shown that the {100} <110> and {112} <110> orientations have the lowest storage energy during the texture evolution [15]. More energy is necessary to activate the nucleation in these zones, corresponding to the complete recrystallization at a higher annealing temperature of 750 °C. Here it is worth mentioning that in the SR sample, the recrystallization fraction curve in Figure 4 shows relative uniform slope, i.e., the recrystallization fraction equably increases with the increase of annealing temperature. This is due to the fact that the {100} <110> and {112} <110> orientations with low storage energy are absent in the SR samples.

### 4.2. Analysis of Texture Evolution after Asymmetric Rolling and Annealing

Figure 12 shows the microstructure and texture at different layers (mainly including S = 0, S = 1/2 and S = 1 layers) after the ASR. EBSD measurements confirm that the {111} <110> and {100} <110>-oriented deformation bands gradually decrease and the {112} <110>-oriented deformation bands gradually increase from the surface to the center layer. This may be related to the stability of the crystal orientation and the shear stress on the material during rolling [34]. In addition, Park and Szpunar [15] calculated stored energy for the α-fiber and γ-fiber components in cold rolled polycrystalline steels and claimed that the stored energy introduced into the deformed specimens is in increasing order for the {100} <110>, {112} <110>, {111} <110> and {111} <112> orientations. This means that deformed {111} <110> and {111} <112> grains should be consumed first and deformed {100} <110> and {112} <110> grains should be last, either by nuclei or by the recrystallized grains, which have already nucleated. This corresponds to our finding that the {111} <110> and {111} <112> orientation strengths are significantly reduced, while the {100} <110> orientation strength remains almost unchanged after annealing at 650 °C, as shown in Figure 7a. In addition, since the recrystallization nucleation preferentially occurs at the shear bands, the orientation of the deformed matrix around the shear bands has a great influence on the orientation of the recrystallized grains. Figure 12 shows that the orientations of the shear bands and the surrounding deformation matrix are very different at different layers. The matrix orientation is {111} <110> at the surface, {112} <110> at the subsurface and {111} <112> and {123} <133> at the center layer. This may be related to the evolution path of grain orientation during cold rolling [39,40]. Related studies have shown that the shear bands embedded in the {112} <110>-oriented deformation matrix mainly provide the nucleation position for the cube- and Goss-oriented recrystallization grains [15], and the shear bands embedded in the {111} <112>- and {111} <110>-oriented deformation matrixes are the preferential nucleation positions of the Goss-oriented recrystallization grains [14,15,40,41,42]. Therefore, the center layer has obvious Goss orientation after annealing at 650 °C, as shown in Figure 11b. However, few Goss-oriented grains are observed at the surface, as shown in Figure 11b, although orientation of the deformed matrix around the shear bands at the surface is {111} <110>. This may be related to the fragmentation phenomenon after cold rolling. Figure 12 shows that the ASR produces severe fragmentation at the surface and subsurface of the sample, while the center layer is relatively bright. The broken small subgrains with different orientations can serve as the core of nucleation during annealing. Therefore, they have the ability to compete with new Goss- and cube-oriented grains. As a result, the growth of Goss- and cube-oriented grains is inhibited to a certain extent at lower annealing temperature. With the increase of annealing temperature, the inhibited Goss- and cube-oriented grains have sufficient energy to grow by swallowing the γ-fiber-oriented grains. Until 750 °C, the γ-fiber-oriented grains at the surface of the sample are almost consumed. Thus, the surface layer of the sample shows a strong η-fiber texture after complete recrystallization, as shown in Figure 8b. However, after annealing at 750 °C, the strength of the γ-fiber texture in the center layer significantly increases, as shown in Figure 8b, staying in good agreement with the literature [4,43]. Studies have shown that at higher temperature, {111} <112>- and {111} <110>-oriented grains easily form large-angle grain boundaries that have a high migration rate with neighboring oriented grains. Then, the large-angle grain boundaries can swallow the surrounding deformed grains (lower energy α-fiber-oriented grains), thereby significantly enhancing the strength of the γ-fiber texture.

## 5. Conclusions

In the present work, evolution of the microstructure and texture of non-oriented 3.3% Si steel after ASR and SR with/without subsequent annealing have been investigated and compared. The conclusions drawn from the results can be summarized as follows:(1)Compared with the SR sample, the shear bands in the ASR sample are more abundant. The shear bands have a larger inclination angle with the RD, generally around 40°. In addition, the surface and sub-surface layers of the ASR sample have obvious grain fragmentation.(2)After annealing at 650 °C, the recrystallization fraction of the ASR sample reaches 62%, which is significantly higher than that of the SR sample (30%). Moreover, when increasing the annealing temperature, the rate of increase of the recrystallization fraction decreases.(3)Compared with the SR, the ASR only changes the strength of the texture components; especially, the strength of the γ-fiber texture is enhanced. No new components are observed. There are also great differences in the evolution of texture in different layers in the ASR sample. After annealing at 650 °C, the nucleation of Goss- and cube-oriented grains is inhibited at the surface layer, while Goss- and cube-oriented grains are nucleated in the center layer. However, after annealing at 750 °C with complete recrystallization, the surface layer completely consists of η-fiber-oriented grains, while the central layer is dominated by the γ-fiber-oriented grains.(4)After annealing at 850 °C, the through-thickness grain orientation in the ASR sample is mainly η-fibers. Both the γ-fibers as the main texture and η-fibers as the subsidiary texture coexist in the SR sample. Thus, the ASR can significantly increase the strength of η-fibers.

## Figures and Tables

**Figure 1 materials-13-04696-f001:**
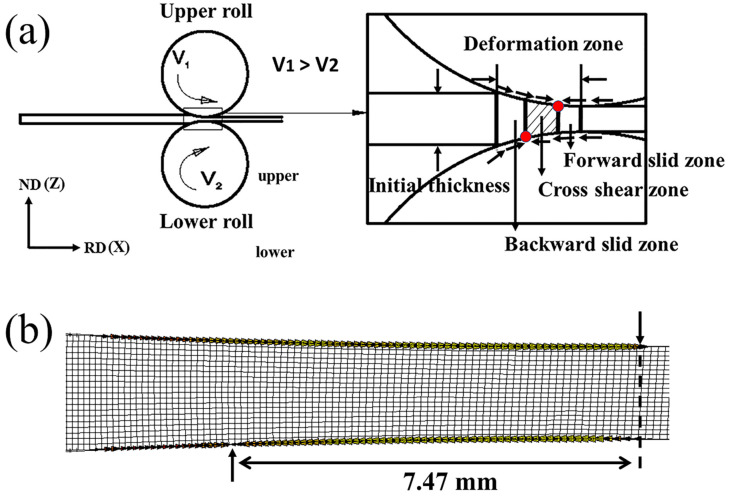
Schematic illustration and the finite element analysis of the asymmetric rolling (ASR). (**a**) Schematic of ASR deformation zone. (**b**) Distortions of rectangular grids predicted by the finite element analysis (FEA). The arrows indicate the locations of the neutral point at the upper and lower surfaces.

**Figure 2 materials-13-04696-f002:**
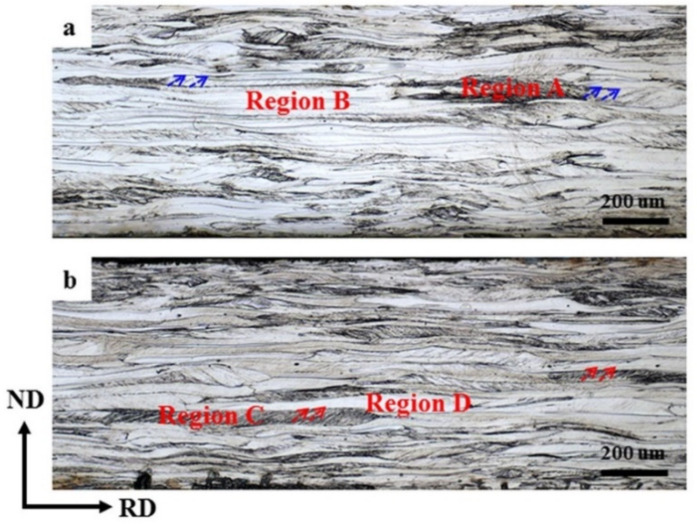
Optical microstructure of sample processed by (**a**) SR and (**b**) ASR. The arrows indicate the shear bands in the samples.

**Figure 3 materials-13-04696-f003:**
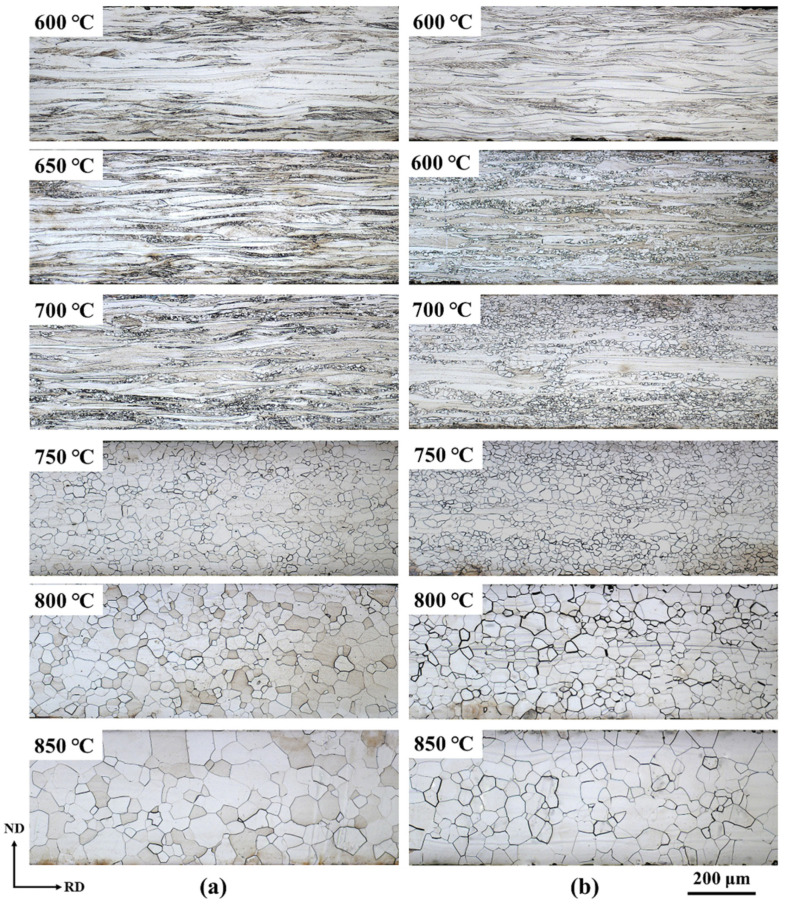
Optical microstructure of samples annealed at different annealing temperatures. (**a**) Symmetric rolling (SR); (**b**) ASR.

**Figure 4 materials-13-04696-f004:**
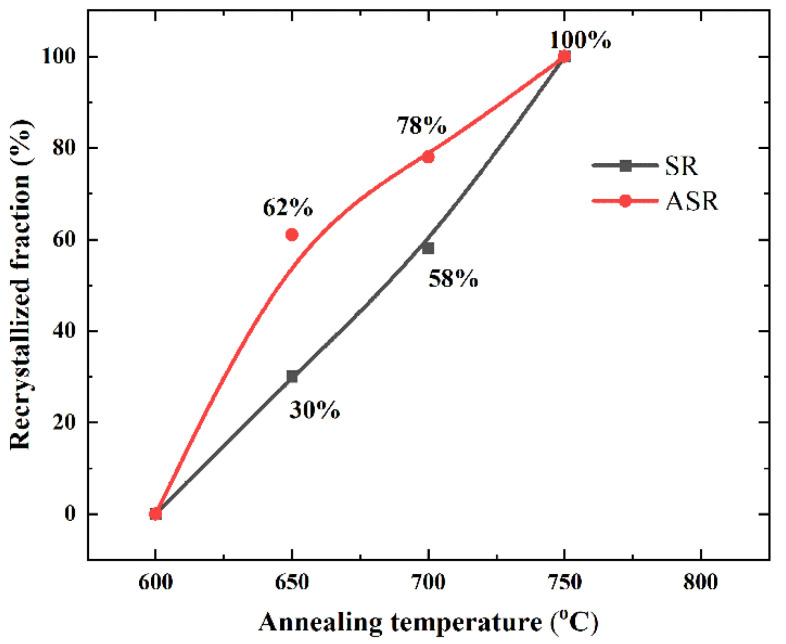
Recrystallized fraction as a function of annealing temperature. The black line is approximated to the nucleation speed at a specific recrystallization temperature.

**Figure 5 materials-13-04696-f005:**
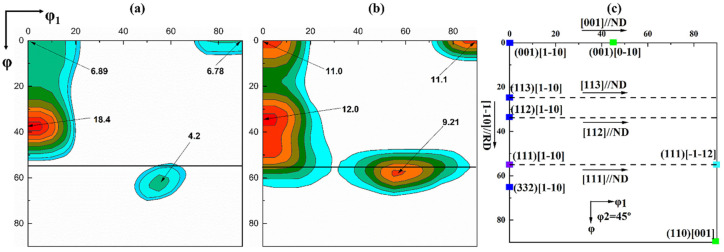
The φ_2_ = 45° section of the orientation distribution function (ODF) obtained from electron backscatter diffraction (EBSD) after cold rolling. (**a**) SR; (**b**) ASR; (**c**) typical texture components and orientation rotation path displayed in the φ_2_ = 45° section of the ODF.

**Figure 6 materials-13-04696-f006:**
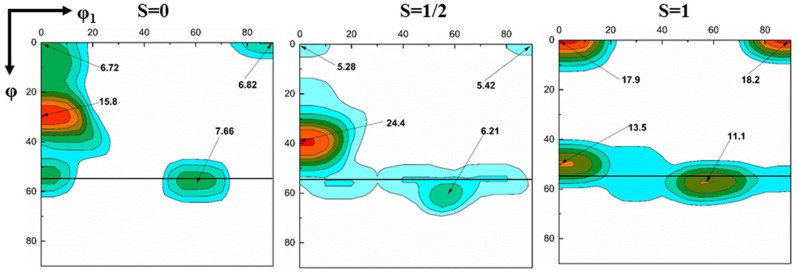
φ_2_ = 45° sections of the ODF at different layers in the ASR cold-rolled sample.

**Figure 7 materials-13-04696-f007:**
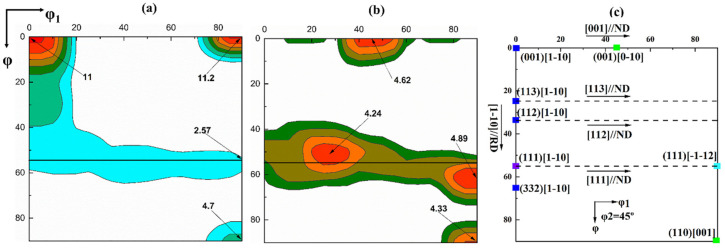
The φ_2_ = 45° section of the ODF obtained from EBSD after annealing (ASR). (**a**) Annealing at 650 °C; (**b**) annealing at 750 °C; (**c**) typical texture components and orientation rotation path displayed in the φ_2_ = 45° section of the ODF.

**Figure 8 materials-13-04696-f008:**
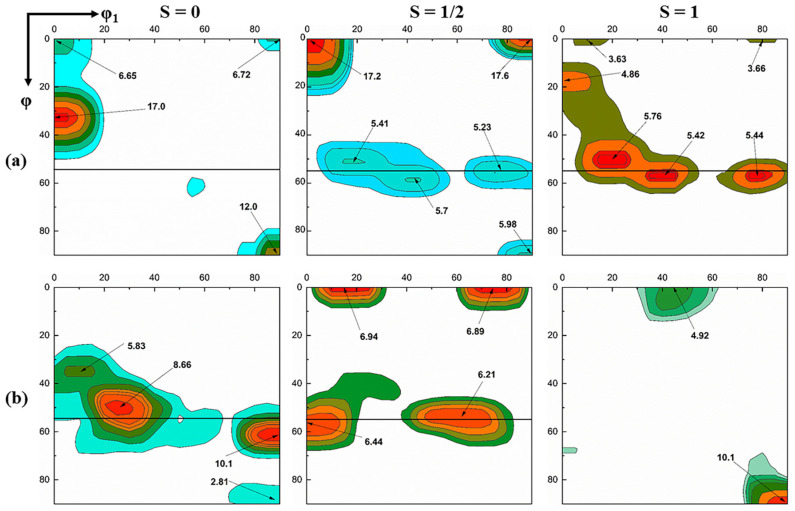
φ_2_ = 45° sections of the ODF at different layers after annealing (ASR). (**a**) Annealing at 650 °C; (**b**): annealing at 750 °C.

**Figure 9 materials-13-04696-f009:**
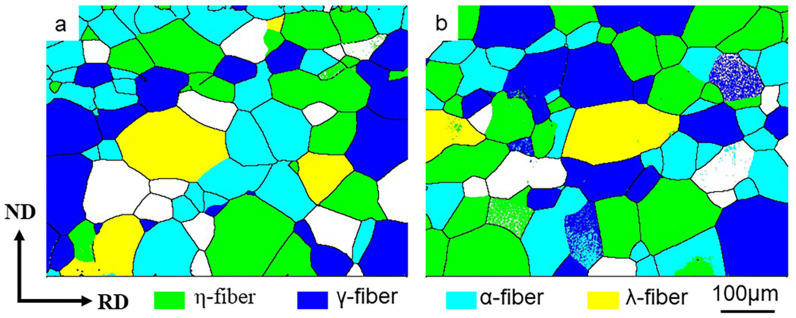
Four typical textures in symmetric-rolled (**a**) and asymmetric-rolled samples (**b**) after annealing at 850 °C.

**Figure 10 materials-13-04696-f010:**
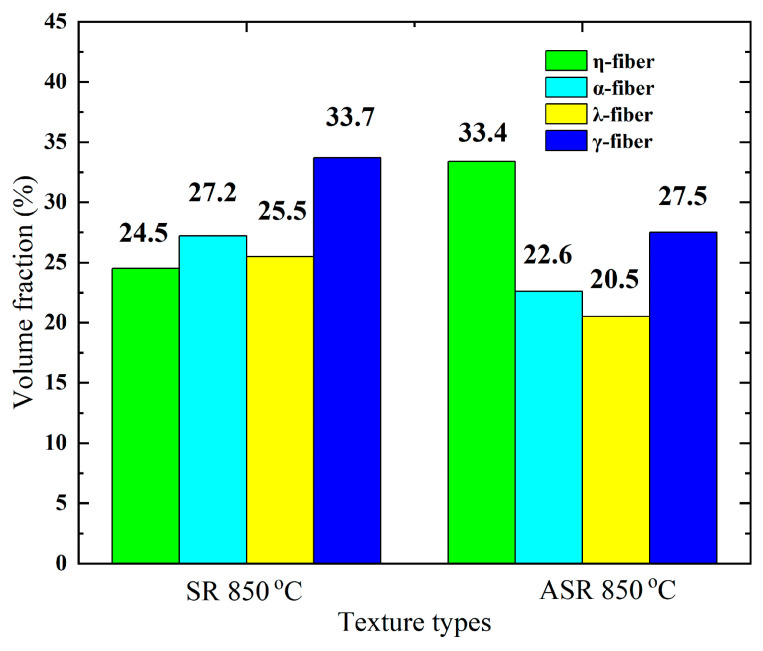
Comparison of four typical texture components after annealing at 850 °C for SR and ASR samples.

**Figure 11 materials-13-04696-f011:**
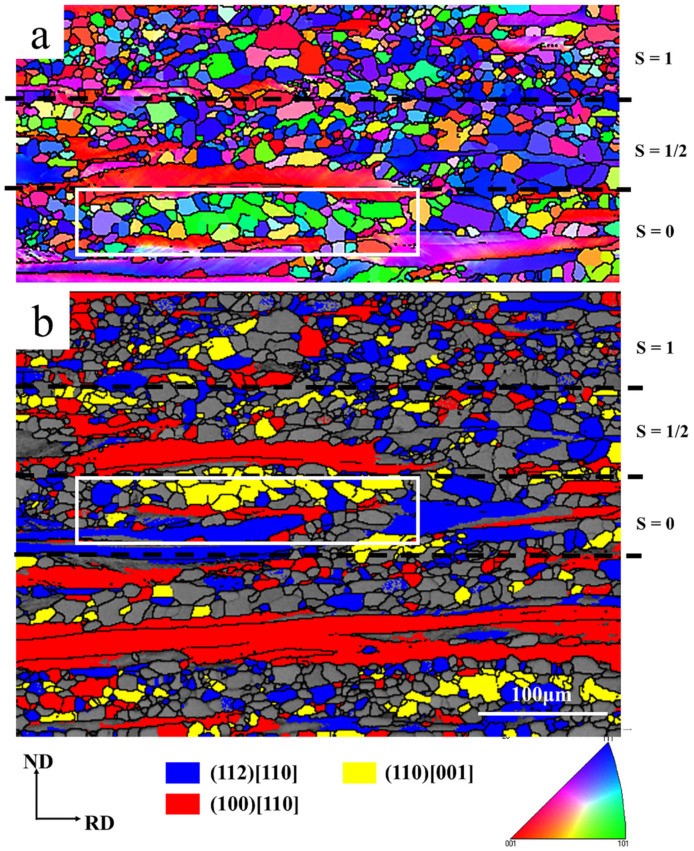
Microstructure and texture of asymmetric-rolled sample after annealing at 650 °C. (**a**) EBSD inverse pole figure map; (**b**) image quality map showing Goss and cube crystals within the deformed matrix.

**Figure 12 materials-13-04696-f012:**
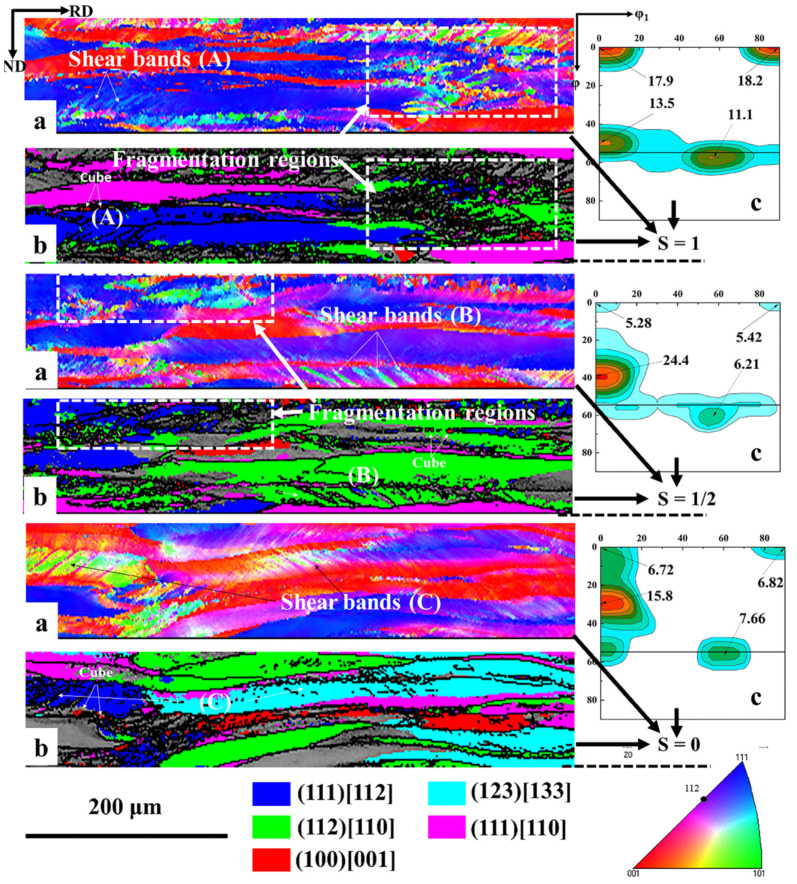
Microstructure and texture after asymmetric rolling. (**a**) EBSD inverse pole figure map; (**b**) image quality map showing orientation within the deformed matrix and shear bands; (**c**) texture (φ_2_ = 45° section, Bunge notation.

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
