# Peer review of "Effect of Shear Bands Induced by Asymmetric Rolling on Microstructure and Texture Evolution of Non-Oriented 3.3% Si Steel"

_materials, 2020, doi:10.3390/ma13214696_

Round 1

Reviewer 1 Report

No reference to [43]. Probably a write error: is [34] supposed to be [43].

Reviewer 2 Report

Review comments

The paper titled “Effect of Shear Bands Induced by Asymmetric Rolling on Microstructure and Texture Evolution of Non-oriented 3.3% Si Steel” investigated the effects of asymmetric rolling on the microstructure and texture of no-orientated Si steel. The results are interesting, but the manuscript in the current version cannot be accepted for publication. I recommend that this article be revised. Some comments as following:

  1. Page 4, “Simultaneously, as the annealing temperature continued to increase, the size of the recrystallized grains generally increased and the grain size distribution gradually became uniform.” Some quantitative analysis data of the grain size and size distribution should be added to support the above claims.
  2. 10, how to calculate “the volume fraction of four typical texture fibers”? Are they the average of different locations? If they are the average of different locations, it is better to add the error bar in the figure.
  3. Page 4, “it is found that the nucleation speed of asymmetric rolled samples continuously decreases with the increase of recrystallization temperature, while symmetric rolled samples remain relatively stable.” This statement seems not accurate. How to define the nucleation speed? All the specimens are annealed at a different temperature from 600-850 oC for 5 minutes, the nucleation fraction increases with annealing temperature increasing, this indicates that the nucleation speed is much faster at a higher temperature. The nucleation kinetics at different temperatures may be different. How about the nucleation speed changes if the anneal time increasing? It is better to add some data of the nucleation fraction of specimens annealed at the same temperature with different time.
  4. The manuscript only shows the results of the rolling speed ratio=1.3 and 1. If the speed ratio is increased, will the shear band increase with the speed ratio increase? What’s the effect of the speed ratio on the recrystallization behavior? If some data about the microstructure and texture evolution of the material after asymmetric rolling with rolling speed ratio, it will help us to understand the effect of the shear band on the recrystallization process comprehensively.

Considering all the problems mentioned here above, some parts of this manuscript should be revised. I recommend that the manuscript should be revised and reviewed again.

Reviewer 3 Report

The authors present a work on shear bands induced by asymmetric rolling and how these shear bands affect the microstructure and texture evolution. The premise of the work is pertinent to the aims of the journal. The authors present a good effort at describing their results and the study. The manuscript is well organized and the discussion is coherent in the later part of the manuscript however, there are some corrections and modifications to the manuscript that are desirable to ensure it is publishable.

1)      Firstly, the abstract needs to be rewritten to clarify the aim, methods and observations described in it. There are typographical errors in the abstract and grammatical errors in Line 5 and Lines 9, 10 and 11 that render it difficult to infer proper meaning from the abstract. This needs to be fixed.

2)      There are some typographical errors along the manuscript and a thorough proof reading is desirable as there are font mismatches. Incorrect reference notations and missing reference citations and so on.

3)      The shear deformation discussed in Section 2 is important to successfully understand the localized effects that are occurring. It would be beneficial to create a simple 2D analysis using analytical or numerical (Finite element) approaches to describe the deformation without having to refer existing works that quote the percentage change or factor change. That would ensure the work is novel and also comprehensive in its description of the methods employed here.

4)      The results in Section 3.1.2 are very interesting and have been summarized well by the authors. It would be interesting to understand the reason behind the reduction in nucleation speed for ASR samples. The authors should be able to provide a little more insight into why these observations would be relevant and what could be the cause when they are comparing ASR and SR samples here. The authors discuss this briefly in section 4.1 but a little more insight into the actual theoretical aspect and quantitative estimation of the migration velocity might merit the entire manuscript.

Round 2

Reviewer 2 Report

The revised manuscript was well revised by the authors based on the reviewer's comments. I have no more comments on the manuscript. I recommend that the manuscript can be accepted for publication in the current version.